# Ontogenetic Development of Sexual Dimorphism in Body Mass of Wild Black-and-White Snub-Nosed Monkey (*Rhinopithecus bieti*)

**DOI:** 10.3390/ani13091576

**Published:** 2023-05-08

**Authors:** Yan-Peng Li, Zhi-Pang Huang, Yin Yang, Xiao-Bin He, Ru-Liang Pan, Xin-Ming He, Gui-Wei Yang, Hua Wu, Liang-Wei Cui, Wen Xiao

**Affiliations:** 1School of Life Sciences, Central China Normal University, Wuhan 430079, China; liyp@eastern-himalaya.cn; 2Institute of Eastern-Himalaya Biodiversity Research, Dali University, Dali 671003, China; 3International Centre of Biodiversity and Primate Conservation, Dali University, Dali 671003, China; 4Institute of International Rivers and Eco-Security, Yunnan University, Kunming 650091, China; 5Administration of Baimaxueshan National Nature Reserve, Diqing 674500, China; 6School of Anatomy, Physiology and Human Biology, The University of Western Australia, Perth WA 6009, Australia; 7Administration of Gaoligongshan National Nature Reserve in Nujiang, Nujiang 673200, China; 8Key Laboratory of Wildlife Conservation for Minimal Population in Universities of Yunnan Province, Southwest Forestry University, Kunming 650224, China

**Keywords:** *Rhinopithecus bieti*, sexual bimaturism, environmental adaptation, multilevel social organization, sexual selection

## Abstract

**Simple Summary:**

Sexual dimorphism widely exists in animals and is reflected in different forms; of these, body mass dimorphism is the most prominent. Studies on the development and adaptation mechanism of sexual dimorphism in body mass can help us to understand how animals adapt to their environment via sexual dimorphism. As this has not been widely reported in *Rhinopithecus*, this study explores the ontogenetic development pattern of sexual dimorphism in the body mass of *R. bieti*, and interprets the causes resulting in extreme sexual dimorphism. The results showed that a significant dimorphism appears when females enter the reproductive period, reaching the maximum when males are mature for reproduction. It was determined that a period of decline begins after 8 years of age, and that males had a longer growth period than females. The large sexual dimorphism in body mass in *R. bieti* can be explained by both Bergmann’s and Rensch’s rules; however, the females’ choice for males may not be significantly related to it. Overall, environmental adaptation, a distinctive alimentary system, and complex social structure have allowed *R. bieti* to have remarkable sexual dimorphism in body mass compared to other colobines. This study will undoubtedly highlight the same issues on the other 26 primate species in China, particularly the colobines, which will enrich research designs and enlarge research focus on China’s primatology.

**Abstract:**

Sexual dimorphism exists widely in animals, manifesting in different forms, such as body size, color, shape, unique characteristics, behavior, and sound. Of these, body mass dimorphism is the most obvious. Studies of evolutionary and ontogenetic development and adaptation mechanisms of animals’ sexual dimorphism in body mass (SDBM), allow us to understand how environment, social group size, diet, and other external factors have driven the selection of sexual dimorphism. There are fewer reports of the ontogenetic development of sexual dimorphism in body mass in *Rhinopithecus*. This study explores the ontogenetic development pattern of SDBM in wild black-and-white snub-nosed monkeys (*R. bieti*), and the causes resulting in extreme sexual dimorphism compared to other colobines. A significant dimorphism with a ratio of 1.27 (*p* < 0.001) appears when females enter the reproductive period around six years old, reaching a peak (1.85, *p* < 0.001) when males become sexually mature. After the age of eight, the SDBM falls to 1.78, but is still significant (*p* < 0.001). The results also indicate that males had a longer body mass growth period than females (8 years vs. 5 years); females in larger breeding units had a significantly higher SDBM than those in smaller ones (2.12 vs. 1.93, *p* < 0.01). A comparative analysis with other colobines further clarifies that *Rhinopithecus* and *Nasalis*, which both have multilevel social organization, have the highest degree of SDBM among all colobines. The large SDBM in *R. bieti* can be explained through Bergman’s and Rensch’s rules. Overall, environmental adaptation, a distinctive alimentary system, and a complex social structure contribute to *R. bieti* having such a remarkable SDBM compared to other colobines. In addition, we found that females’ choice for males may not be significantly related to the development of SDBM.

## 1. Introduction

Sexual dimorphism—where the two sexes of the same species differ in external appearance or in other ways—exists extensively in animals [1], such as amphibians [2], reptiles [3], birds [4], and mammals [5]. Sexual dimorphism is displayed in several ways, such as body size (mass), color, shape, behavior, and sound [6]. For example, adult males and females have different coloring in most gibbon species [7], and ungulate males feature fully developed antlers [8,9]. Sexual dimorphism is an important physiological characteristic for understanding the adaptation of animals to the natural and societal environment [10]. Many studies have focused on it, as early as Darwin’s theory of evolution [11,12,13,14], and it is considered to be the consequence of natural adaptation and sexual selection [15,16,17,18].

Sexual dimorphism is also a pervasive phenomenon in primates [19] and may take several forms [15,20], mainly shown in body mass, canine tooth size, sex skin, and color, but has most commonly been described as sex differences in body mass [21,22], which can be measured reliably [23]. The past several decades have seen a steady growth in studies of sexual dimorphism, including simple documentation of the phenomenon in various species and broad comparative analyses in a wide variety of primates. Consequently, sexual selection is considered to be the most influential theory that explains sexual dimorphism in body mass [24], which can be caused by male–male competition for females or female mates selection [4,10,11,14]. Some studies further show that the sexual dimorphism in body size (male vs. female) is more significant in polygynous than in monogamous species [25,26,27,28]. Males face considerable social pressures for mating competition, so adult males are more muscular than females [17,29], especially in species with a multilevel society [30,31,32]. In addition, natural selection states that animals living at higher altitudes, and hence facing severe resource competition and lower temperate challenges, have a larger body size, and maintain more prominent sexual dimorphism [33,34]. Sexual bimaturism has also been widely thought to explain sexual dimorphism in primates [35].

Black-and-white snub-nosed monkeys (*Rhinopithecus bieti*) belong to the Cercopithecidae family and colobine sub-family. They are endemic to China and are distributed in the upper reaches of the Mekong and the Yangtze Rivers, where the Qinghai-Tibet and Yun-Gui Plateaus meet [36]. Their southern-most range is located in Yunlong County, Yunnan, while the northern-most range is in Mangkang County, Tibet [36,37,38]. They are the non-human primate species which distributed and adapted to the highest altitude, and also are a flagship taxon of the cold temperate coniferous forest ecosystem in the northern hemisphere [36] Asian colobines initially originated from Africa in the Middle Miocene, migrating into Eurasia through the gateway of North Africa, possibly in the Late Miocene [39,40]. The ancestor of the odd-nosed forms (*Rhinopithecus*, *Nasalis*, *Simias*, and *Pygathrix*) has been unearthed recently in Zhaotong, Yunnan, in the deposit of the Late Miocene or Early Pliocene [39,41,42]. The Himalayan group (*R. bieti* and *R. strykeri*) diverged from the other snub-nosed species (*R. roxellana* and *R. brelichi*) around ~1.99 Mya according to phylogenetic analyses [43], following which the *R. strykeri* and the *R. bieti* diverged around 0.60 Mya. Of these, *R. bieti* is unique, remaining in the Qinghai-Tibet Plateaus and Hengduan mountains that have been significantly modified due to the accelerated tectonic uplift of the plateaus and mountains, after settling down in the places where they are residing [29,44]. Others in *Nasalis* and *Pygathrix* moved south along the valleys and riverbanks of the Mekong and Salween to disperse into Southeast Asia [45,46]. In other words, *R. bieti* is distinct due to its habitat at the highest elevation and the harshest environment with limited distribution areas confined by the Mekong and Yangtze Rivers [36,37,47] (Figure 1). Males and females are sexually mature when they are five years old, but while females can enter the breeding system straight away, males are usually active around eight years old [29,36]. The species is a typical primate taxon featured by a multilevel social system [48]. Its polygamous mating system consists of multiple reproductive units; there are several OMUs (one adult male and several females unit) and one or more AMU (all males unit) in one natural population, and the males have different reproductive opportunities between the two kinds of units [49]. Such a social structure and mating system leads to fierce male reproductive competition [36]. The largest adult male/female ratio in the breeding unit is 1:5 [50,51], an indication of extensive sexual competition among males and females, which may stimulate males to improve their reproductive capacity by increasing their body mass [10]. Thus, this species is an ideal model for discussing adaptation and sexual selection by studying sexual dimorphism in body mass (SDBM) in colobines.

Studies of evolutionary and ontogenetic development and adaptation mechanisms of animals’ sexual dimorphism in body mass (SDBM) will increase our understanding of how environment, social group size, diet, and other external factors have driven the selection of sexual dimorphism [5,52,53]. This study will help clarify whether the specific environments, habitats, and climate adopted by the *R. bieti* have shaped the SDBM of this species quite differently compared with other colobines. It will also be fascinating to know whether the ontogenetic development of such dimorphism, which is difficult (if not impossible) to measure in wild animals, differs from those reported for other primate species. Thus, we will describe the ontogenetic development of sexual dimorphism in body mass of wild black-and-white snub-nosed monkeys via the continuous and accurately recorded data of body mass. Then, to discuss the genesis of sexual dimorphism in body mass of these monkeys (*R. bieti*), our study will tackle the following three questions in *R. bieti*: (1) can we define an age threshold for SDBM appearance during the ontogenesis; (2) can we explain the significant body mass and SDBM of *R. bieti* by Bergmann’s and Rensch’s rules; and (3) does a significant SDBM exist in larger grouping populations due to more extensive reproductive competition pressures.

## 2. Materials and Methods

### 2.1. Study Area and Object

The research station is located in Shangri-La Yunnan Golden Monkey National Park, China (27°30′ N, 99°20′ E) where the altitude ranges from 2500 to 3800 m. The area is primarily covered by coniferous and broad-leaved forests [54]. The average annual temperature is 9.4 °C, the minimum is −6.0 °C, and annual precipitation is 1200 mm (Smart Weather Stations in the study area, a model of HOBO RG-3M).

This study is part of a long-term research project on the behavior and ecology of *R. bieti* in Baimaxueshan National Nature Reserve, Weixi County, Yunnan province, China. The focal population consists of 87 individuals belonging to eight OMUs (one male unit) and one AMU (all male unit), sized from 3 to 13 per unit (on 15 April 2022, see Table 1). They were classified into four age subgroups: infant, juvenile, adult male (age ≥ 8 years old), and adult female (age ≥ 5 years old) [35]. The number of adult females per OMU varied from 1 to 7 (Table 1). The structure and individual number of units can change at any time; female individuals rarely migrate between breeding units, while male individuals must leave the breeding unit and enter the all-male unit at 3–4 years old. There was a large population (around 500 individuals) nearby our focal population, meaning that individual migration between groups was a frequent occurrence.

### 2.2. Individual Identification

We identified the OMUs via differences in member composition and lead males. For example, there were 10 individuals in the DG OMU, comprising 2 infants, 3 juveniles, 4 adult females, and 1 adult male, and there were also 10 individuals in the YDH OMU, but comprising 1 infant, 4 juveniles, 4 adult females, and 1 adult male, on 15th March 2013 (Table 1).

Individuals were identified by obvious, easy to distinguish, and relatively stable characteristics; for example, lip markings, special coat color on the back and chest, facial scars, deformity of limbs, and different body characteristics were used for individual identification (Figure 2). Monkeys had been well habituated to the presence of the researchers after many years of food provision and monitoring. Thus, we could observe them from a distance of less than 20 m.

### 2.3. Body Mass Measuring

Body mass was measured using a wireless electronic scale (XK3190-A12-E, YaoHua, ShangHai, China, max value 100.00 kg, min value 0.02 kg) designed for an environment with a relative humidity of 10%~85% and a temperature range between −10 °C and 40 °C (Figure 3).

The body mass recording was carried out between January 2010 and March 2013, in which the data of 996 body mass measurements were collected. We spent more than five days each month recording body mass data of individuals in the focal population. We placed a wireless electronic scale on a relatively flat place in the feeding site before the monkeys went into the feeding site in the afternoon. We could obtain the body mass data when a single monkey would go to eat the foods that had been placed on the scale (the stability light of the scale must display on). Meanwhile, we could also quickly identify the individual and record its age, gender, unit, and other information, as illustrated in Figure 3. Some individuals in a given group were weighed more often than others, but we only recorded once per individual per month. However, this does not influence the validity of the statistical application, since the average values of the age-sex class, instead of individual records, were used in the comparison.

### 2.4. Body Mass of Species in Colobine

For comparison, we collected adult body mass (mean value), species’ social group size, and distribution parameters (median latitude and altitude) of ten colobine genera at species level found in Africa and Asia, using data from the data sources (see the Table 2). The geographic range of each species of colobines was download from the IUCN red list (https://www.iucnredlist.org/, last accessed on 15 January 2023), and the median values of altitude and latitude were obtained using Qgis (https://www.qgis.org/en/site/, last accessed on 10 February 2023).

### 2.5. Statistical Analysis

The SDBM was gauged with a ratio of male body mass/female body mass, and the ontogenetic development was analyzed for individuals one to eight years old, and for individuals older than eight. The annual increments in body mass were gauged with a D-value between two years, and the annual increment rate in body mass was gauged with a ratio of body mass D-value/body mass in the first year. The average sex ratio (male/female, M/F) of each OMU in the focal breeding band was 3.4; therefore, we set 4 mates as a critical level for OMU size, called breeding units with 5 or more adult females (as the mate for the alpha male) larger OMUs, and breeding units with 4 or fewer adult females smaller OMUs. Then, we compared SDBM between OMUs with ≤4 and ≥5 adult females. This choice is based on the reasoning that the fewer females per OMU, the less significant reproductive competition, and that the larger the SDBM, the stronger the sexual selection tends to be [21].

For determining samples, one individual monkey over a four-year period can be considered to be four independent samples, because of its advancing age. For example, the monkey called LingXing was born in 2010, so it was assigned to the one-year group, then to the two-year group in 2011, and so on. We measured the body mass of each individual per month, so that we obtained data from every individual several times per year, but we calculated one average value of the several pieces of data from one individual as one sample of every individual per year. The body mass yearly average can effectively avoid the impact of the seasonal fluctuation on results; because of obvious seasonal fluctuations in the body mass in *R. bieti*, the body mass of *R. bieti* varied significantly: 3.56 kg (15.43%) for adult males and 1.69 kg (14.08%) for adult females in one year.

We used an Independent Sample Test to analyze differences in body mass among age groups, between males and females, and analyze differences in body mass and SDBM between lager OMUs and smaller OMUs. The data which correspond to normal distribution were analyzed using *t*-tests, while the Mann–Whitney test was used if the data did not correspond to a normal distribution. We used a linear model of correlation analysis to explore the relationship between the body mass or SDBM and the species’ social group size. We also studied such a relationship with the median values of altitude and latitude of each colobine species separately.

Statistical analyses were performed with R Statistics version 1.1.442, Origin Pro 8.0 software (https://www.originlab.com/origin, last accessed on 10 February 2023). Maps were drawn using Qgis (https://www.qgis.org/en/site/, last accessed on 10 February 2023).

## 3. Results

### 3.1. Ontogenetic Development of SDBM in R. bieti

The highest annual increase in body mass was 2.3 kg, measured in the four to five-year-old females. Female body mass increased rapidly from birth to four years old (the annual increment in body mass was 1.9 kg, and the annual increment rate in body mass was 65.71%), then went into a slow growth period from five to seven years old (annual increment in body mass was 0.9 kg, annual increment rate in body mass was 9.65%); female body mass stopped growing after seven years old (annual increment in body mass was −0.1 kg, annual increment rate in body mass was −0.10%) (Figure 4). In males the highest annual increase in body mass was 4.8 kg, recorded in the seven to eight-year-olds. There were two body mass growth periods: the most rapid growth period for the males’ body mass was from birth to four years old (annual increment in body mass was 1.9 kg, annual increment rate in body mass was 56.67%); the second most rapid growth period was from five to seven years old (annual increment in body mass was 4.2 kg, annual increment rate in body mass was 33.28%), and male body mass also stopped growth after seven years old (Figure 4). Thus, for males, there was a longer body mass rapid growth period (birth to seven years old) compared with females (birth to four years old) in *R. bieti* (Figure 4).

As for adults, the body mass of males 8 years or older was 21.5 ± 1.8 kg (N = 131, range 17.6–25.8). Females 5 years or older had a body mass of 11.3 ± 1.4 kg (N = 323, range 7.2–14.2). These measurements indicate a significant body mass difference between males and females (N_male_ = 131, N_female_ = 323, df = 16.70, *p* < 0.001), and the SDBM was 1.90.

The SDBM within the first year was slightly different (1.21), reaching a significant level (SDBM = 1.21, body mass male 1.7 kg ± 0.6 kg vs. female 1.4 ± 0.5 kg, N_male_ = 35, N_female_ = 36, t = 2.16, df = 69, *p* < 0.05), but there is no significant difference for the 2 to 5-year-old groups (Figure 5). Another wave of significant SDBM increase appears when monkeys are more than 6 years old (SDBM = 1.27, body mass male 12.6 kg ± 3.0 kg vs. female 9.9 ± 0.7 kg, N_male_ = 25, N_female_ = 70, Z = 4.81, *p* < 0.001), reaching the peak after 8 years (SDBM = 1.85, body mass male 21.8 kg ± 2.2 kg vs. female 11.8 ± 1.2 kg, N_male_ = 35, N_female_ = 116, Z = 8.952, *p* < 0.001). After 8 years the SDBM fell to 1.78, but it was still significant (body mass male 21.3 kg ± 1.5 kg vs. female 12.0 ± 0.6 kg, N_male_ = 96, N_female_ = 112, t = 47.71, df = 206, *p* < 0.001) (Figure 5).

### 3.2. The Relationship between Body Mass, SDBM, and OMUs Sizes

Adult male body mass in the larger unit was not significantly higher than those from the smaller units (female/male ≥ 4), 21.9 ± SD 1.3 kg vs. 21.4 ± SD 1.1 kg (N_larger_ = 12, N_smaller_ = 13, t = 0.93, df = 23, *p* = 0.361), but the adult females’ body mass in the larger OMU was a little less than in the smaller OMUs, 10.6 ± SD 1.65 kg vs. 11.4 ± SD 1.3 kg (N_larger_ = 61, N_smaller_ = 32, t = 2.30, df = 91, *p* = 0.024) (Figure 6A).

A significant difference in SDBM existed between the two groups (Figure 6B), with the larger OMUs (adult females ≥ 5) showing greater dimorphism than smaller OMUs (adult females ≤ 4), 2.12 ± SD 0.36 vs. 1.93 ± SD 0.25 (N_larger_ = 61, N_smaller_ = 32, Z = 2.65, *p* = 0.008).

### 3.3. Sexual Dimorphism in Body Mass Comparison among Colobine genera

The mean body mass of *Rhinopithecus* was 16.9 kg and 9.4 kg for adult males and females, respectively. The SDBM was 1.80, which is slightly smaller than that of *Nasalis* (1.91), but larger than all other analyzed genera (Table 2). The body mass we recorded for *R. bieti* is the highest body mass (male: the maximum was 25.8 kg, and the mean value was 21.5 kg; female: the maximum was 14.2 kg, and the mean value was 11.3 kg) among the colobines (Table 2).

The adult male body mass of colobines (the species of each genus in Table 2 was included in the analysis) was found to be positively correlated with the species’ social group size (N = 51, r = 0.535, and *p* < 0.001) (Figure 7M-1), but not for females (N = 51, r = 0.243, and *p* = 0.086) (Figure 7F-1). Body mass was also positively correlated with the median latitude of their geographic distribution (males, N = 51, r = 0.561, and *p* < 0.001; females, N = 51, r = 0.537, and *p* < 0.001) (Figure 7M-2,F-2), as well as altitude (males, N = 51, r = 0.559, and *p* < 0.001; females, N = 51, r = 0.456, and *p* < 0.001) (Figure 7M-3,F-3).

According to the variation in SDBM among colobines (Figure 8A), we can find that the largest SDBM (1.91) can be seen in *Nasalis*, followed by *Rhinopithecus* (1.80), while *Presbytis* has the smallest (1.09). The odd-nosed group (*Pygathrix*, *Rhinopithecus*, *Simias*, and *Nasalis*) has a higher SDBM than the others (Figure 8A). The figures also presented the relationships between SDBM, species’ social group size, and distribution parameters. SDBM is significantly positively related to body mass (males’ body mass: N = 51, r = 0.784, *p* < 0.001; females’ body mass: N = 51, r = 0.422, *p* = 0.002), species’ social group size (N = 51, r = 0.724, and *p* < 0.001) (Figure 8B-1), median latitude (N = 51, r = 0.413, and *p* = 0.003) (Figure 8B-2), and median altitude (N = 51, r = 0.485, and *p* < 0.001) (Figure 8B-3).

## 4. Discussion

This study aimed to analyze the ontogenetic development of body mass and sexual dimorphism in body mass (SDBM) of the black-and-white snub-nosed monkey (*R. bieti*), a primate species occurring at the highest elevation. Six years was the threshold age for SDBM to appear during the ontogenesis in *R. bieti*. Such significant sexual dimorphism in body mass in *R. bieti* can be well explained by Bergmann’s and Rensch’s rules rather than sexual selection.

### 4.1. Sexual Bimaturism of R. bieti

Sexual bimaturism indicates sexual dimorphism in the ontogenetic development of the same species [79]. The extended growth phase of male primates generally leads them to be larger than females [80], which can also lead to a significant sexual dimorphism in adults [35,81]. Helping us to understand the ontogenetic patterns of sexual dimorphism is critical to explore growth pattern variations among species. Unfortunately, such studies of wild primates are still rare, due to the difficulty in obtaining timely records. A few studies include research on orangutans (*Pongo abelii*) that reach sexual maturity around 15 years old, with males having a longer growth period than females [82]. Our findings (Figure 4) showed that the body mass growth rate declined rapidly for females over four years old, meaning that the female rapid growth period was just four years, but for males the rapid growth period was eight years; thus, sexual bimaturism can explain the difference in body mass of *R. bieti*. This time series of comparisons of sexual dimorphism from infant to adults, as recorded in this study, is not available for other nonhuman primates.

*R. bieti* showed the first signs of SDBM when they were one year old (1.21—males being heavier than females), representing a first wave of significant sexual dimorphism. The results showed the same growth rate for males and females between the ages of two and five, so that the SDBM was not significantly different during this period. The SDBM progressively increased after entering the six years’ phase, and reached a peak after eight years (Figure 5). We propose that the first wave of SDBM of *R. bieti* within the first year may be caused by gender preference of parental investment. Such a phenomenon is probably associated with the unbalanced investment from parents in caring for offspring in species with a polygamous mating system [83], in which infants’ food during the early period of life is mainly breast milk [84,85,86], so we surmise that the rapid growth of males is derived from parents’ investment preference.

The annual increment in body mass took a brief downward trend after the high-intensity parental investment period (one to two years old). The ontogenetic development of body mass restarted again in three or four years old, when most males started to leave the breeding units to join the all-male units, and females remained in the breeding units [50,87]. The body mass growth rate was same for males and females at this time, so that the SDBM was not significant. The fact that a significant SDBM occurred after six years must be associated with males’ longer developmental period. The body mass of males continued to rapidly increase after six years, while the growth rate of females’ body mass decreased rapidly (Figure 4). Thus, the period between six and eight years old was critical for *R. bieti* males, when they needed to increase their physical fitness and experience for the potential challenge to OMU leaders. They obtain the most significant sexual dimorphism at eight years old, meaning that these males have the largest body mass necessary to prepare for the difficult challenges for territory, leadership, and mating opportunities [88].

### 4.2. Adult Sexual Dimorphism in Body Mass in R. bieti

*Rhinopithecus* is the colobine with the most significant body mass and SDBM except *Nasalis* (Figure 7 and Figure 8). *Rhinopithecus* live at higher altitude and latitude areas, and have larger society group sizes that other colobine genera. This can be explained by Bergmann’s rule [34] that finds that *Rhinopithecus* have the largest body mass, and also accommodates Rensch’s rule, that sexual dimorphism in body mass is more pronounced in the species with a larger body mass (males’ body mass: N = 51, r = 0.784, *p* < 0.001; females’ body mass: N = 51, r = 0.422, *p* < 0.01).

Body mass and SDBM in *R. bieti* and other colobines show significant relationships with latitude and altitude (Figure 7 and Figure 8). Both increase according to increasing altitude or increasing latitude from the equator. Such a phenomenon clarifies Bergmann’s rule. However, the unique position of *R. bieti*, located far above the regression line, indicates that, relative to body mass, this species has the most significant society group size (F-1 and M-1 in Figure 7), and inhabits the highest altitudes (F-3 and M-3 in Figure 7). The same scenarios were found when looking at the SDBM (B-1 and B-3 in Figure 8). While all colobine species were studied simultaneously with correlation analysis to figure out their characteristics, *R. bieti* is singled out from the others due to its altitude and social group size, which, as discussed above, led to its unique body mass and SDBM pattern. Within the *Rhinopithecus* genus, *R. bieti* has the most significant SDBM (1.90); *R. brelichi* has 1.88 [51,89], 1.81 for *R. roxellana* [75], 1.75 in *R. avunculus* [51], and 1.65 in *R. strykeri* [76]. Although *R. bieti*’s distribution latitude is not as extensive as that of *R. roxellana*, it is still found in the highest altitude areas, and also has the largest social group size, making a significant contribution to body mass and SDBM. A larger body mass in *R. bieti* has evolved according to natural selection and environmental adaptation [90].

Moreover, besides what has been explained by Bergmann’s and Rensch’s rules, the unique SDBM in *R. bieti* may be closely related to its unique digestive system and dietary selection. It is recognized that colobines that rely on very fibrous foods as a fallback, which are accommodated by their quadripartite stomachs, allow them to have large body size and more developed digestive systems, in contrast to colobines that are fruit- and seed-eating species [91]. *R. bieti* relies heavily on fibrous lichens and bark (>95% feeding time) as fallback foods at elevations of 2625 to 4600 m in winter [36,48,92]. It has a much more developed digestive system—more extensive and more prolonged, especially compared with other low-altitude colobines—and this promotes the efficient absorption of highly fibrous materials [93,94,95,96,97]. Thus, the larger body size in *R. bieti* allows the species to store more reserved substances and consume less energy for effective thermal regulation. The high ratio of body volume to body surface area ensures maximum metabolic heat retention. Such a mechanism helps the species survive on low quality foods during long freezing periods in winter [33,75,98,99].

Special multilevel social organization will likely accelerate these trends. Compared with other colobines, *Rhinopithecus* (1.80) and *Nasalis* (1.91) have a bigger SDBM, and they also have an extremely complex and multilevel social organization with a stricter hierarchical structure [49,100], increasing fierce intraspecies competition so that individuals have to increase body mass and SDBM [5,21,101]. Using body mass data from 37 captive primates, Leigh showed that species with low levels of male –male competition (monogamous/polygamous mating systems) exhibit fewer sex differences in development [80,102]. In colobines, the basic social group among different species is one male and multi-female units, and polygyny is the common mating system for a large proportion of colobine species, such as *Simias*, *Presbytis*, *Nasalis*, and *Rhinopithecus* [28,48]. Accordingly, those polygynous species should have a high degree of SDBM due to the high male–male competition [19,103]. However, many island or peninsula *Presbytis* species, such as *P. hosei* (5.8–6.5 kg, M/F = 1.13), *P. canicrus* (5.5–7 kg, M/F = 1.13), and *P. melalophos* (5.2–9 kg, and M/F = 1.06), have small body mass and a small degree of SDBM (mean 1.03 for all species) (see Table 2). Although no reports of allometric growth exist for these species, some studies show that both sexes of *Presbytis* (*P. entellus*, *P. rubicunda*, *P. sabana*) have rapid development and reach maturity and fertility around the age of 3 years [55,104]. Their small body size and sexual dimorphism seem to be caused by similar growth rates and how island lineages evolve (limited resources limiting body size and energy acquisition and thus leading to island dwarfism) [105]. For other colobines, SDBM begins to emerge and increase along the mating system spectrum, from polygynandrous to polygynous and social organization, such as 1.16 for *Trachypithecus* (e.g., *T. cristatus* is 1.16, *T. leucocephalus* is 1.19, and *T. phayrei* is 1.06), 1.33 for *Simias* (one-male grouping, *S. concolor* is 1.33), 1.31 for *Semnopithecus* (multi-male multi-female grouping, e.g., *S. hector* is 1.35, *S. hypoleucos* is 1.23, and *S. priam* is 1.36), and 1.33 for *Pygathrix* (semi-multilevel grouping, e.g., *P. cinerea* is 1.25, *P. nemaeus* is 1.40, and *P. nigripes* 1.34) [28,78,89]. What’s more, *Rhinopithecus* (1.80) and *Nasalis* (1.91) with multilevel social organization have much higher SDBM than other colobines.

Another factor resulting in a larger SDBM in *R. bieti* may be the preemption for the males, especially regarding higher quality food [106], which has been reported in other mammal species [107]. In other words, males usually obtain more nutritional components to increase body size for alternative pressure-driving selection purposes. In addition, the OMU male also faces challenges from outside his group; one or two single males habitually stay nearby the OMUs, yielding a significant threat to his group, especially the prospect of taking over the group [87,108]. Thus, the reproductive competition in *R. bieti* males is very intense among OMUs, and between OMUs and AMUs. We recorded two serious fighting events in which the male leader in an OMU was expelled by another male, and finally died (Figure 9). Such fierce pressure from reproductive competition also encourages males to increase their reproductive competitiveness by gaining body mas, leading to an increased degree of SDBM [16].

However, the body mass and SDBM of *Nasalis* colobines can not only be explained using the above reasoning. According to Roos et al., after leaving Africa for Asia in the middle Miocene (10–11 Mya), colobines may have arrived at a convergence –divergence center (CDC) consisting of the Qinghai-Tibet Plateau and Hengduan mountains [46,109] during the Late Miocene (about 8.5 Mya) or later Pliocene, where related fossils, such as *Rhinopithecus*, *Trachypithecus* in Tongzigou [110,111], and *Mesopithecus* from Zhaotong [112], have been discovered. They then successfully used the geographic structures of the CDC to complete their dispersion through East Asia, resulting in *Rhinopithecus* fossils being found in mainland China and Taiwan [113,114], and *Dolichopithecus* (*Kanagawapithecus*) in Japan [115]. A western dispersion from the CDC is supported by the fossil *Semnopithecus* from the late Pleistocene found in India [114], and Myanmar (*Semnopithecus*) from the late Pleistocene [116], as well as fossils from another recently recorded species, *Myanmarcolobus yawensis*, from the early Pliocene found in Myanmar [117]. A southward dispersion of the colobines from the CDC can be confirmed by the fossils from ancestors of the existing odd-nosed monkeys (*Nasalis*) and *Trachypithecus*/*Presbytis* in Southeast Asia, such as *Trachypithecus auratus* found in Java in the middle Pleistocene [46,118]. Their ancestor once encountered the Hengduan Mountains (8.0–6.0 Mya) during a glacial period in the late Miocene, so that it adapted to the cold climate, resulting in a larger body size, then migrated southward and crossed the Indo-China Peninsula to enter Borneo at approximately 6.5 (7.0–5.7) Mya (Qi et al., accepted). Thus, it seems that larger body size and SDBM in odd-nosed monkeys (*Pygathrix*, *Rhinopithecus*, *Simias*, and *Nasalis*), in contrast to their African and Asian counterparts, are closely related to their shared ancestors who experienced evolutionary development on the Qinghai-Tibet Plateaus and the Hengduan mountains during the Quaternary glaciation period.

The fact that *Nasalis larvatus* has the most significant body mass and SDBM among the colobines may be related to its evolutionary development addressed above. It is also possible for it to have undergone unusual phylogenetic development, referring to its unique morphology, a distinct taxon from the other odd-nosed taxa in colobines. Thus, we propose that *Nasalis* has possibly kept the original physical characteristics in its phylogenetic development after separating from the odd-nosed monkey, such as nasal structure. In addition, other selective forces may have resulted in the very large body size and extreme SDBM of *Nasalis*, such as the multilevel social organization that has been described.

### 4.3. The Relationship between SDBM and OMU Size

Our findings (Figure 6A) illustrate a significant relationship between body mass and OMU size in females but not males. The females in smaller OMUs had a higher body mass than those in larger OMUs. This may be because inter-group competition for natural resources or reproductive opportunities is greater among females in larger OMUs. This result is also in accordance with the fact that colobine species have greater body mass, and larger social group sizes (Figure 7F-1). In other words, larger groups of animals usually face more substantial pressure for food resources, especially during cold and dry winters where food is scarce [6,21,36,75,119].

We also showed a significant differentiation between SDBM and OMU size (Figure 6B); a greater SDBM accompanies a larger group size, which implies that SDBM is tightly related to the degree of selection pressure within a social or group structure: the larger the social or breeding unit size, the more severe the selection pressure from the territorial domain, food resources, and mating opportunities [5,101]. Thus, a greater SDBM in a larger group may be the result of alpha males being able to monopolize these small groups of females and benefit from associating with these units on a long-term basis so as to be able to reliably track female reproductive condition, and prevent females from mating promiscuously [49,100].

### 4.4. The SDBM Cannot Be Explained Forcefully by Sexual Selection in R. bieti

This study also indicates that SDBM is not closely related to female choice in *R. bieti*. What is illustrated in Figure 6A suggests that although SDBM was significantly different between smaller and larger OMUs, males’ body mass in larger OMUs was not significantly higher than in smaller OMUs. This notion is also supported by behaviors observed in *R. roxellana*. Specifically, females in *R. roxellana* did not automatically choose the male challenger who overpowered the resident male, suggesting that male fighting ability has little effect on the OMU formation and females’ mate selection [120].

## 5. Conclusions

The fierce environmental pressures, such as the high altitude, cold climate, unique diet, and complex social structure addressed above have prompted *R. bieti* to develop an enormous SDBM compared to other colobines. The significant ontogenetic development in SDBM begins at year five, when females enter the breeding system, and reaches a maximum at eight years old when the males are ready for mating competition. In addition, the females’ choice of a male may not be significantly related to the formation of SDBM in *R. bieti*.

## Figures and Tables

**Figure 1 animals-13-01576-f001:**
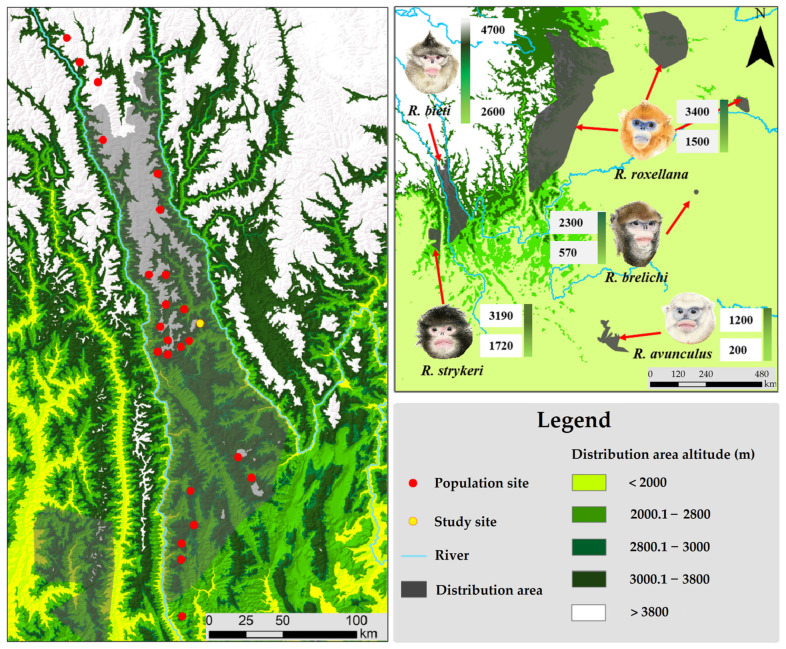
Distribution areas of the *Rhinopithecus* species and the population distribution sites of *Rhinopithecus bieti.*

**Figure 2 animals-13-01576-f002:**
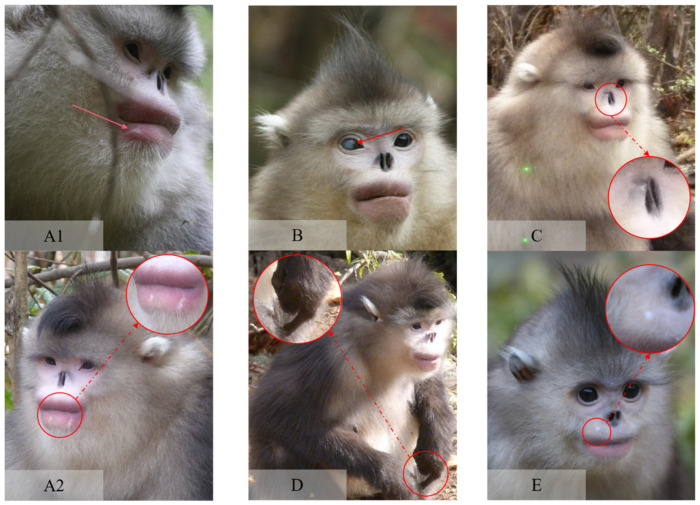
Some visible body, face, and eye characteristics used for individual identification, besides pelage colors that vary between age groups and individuals. (**A1**,**A2**): Lip scars on different locations with various numbers; (**B**): eye marker caused by corneal injury or cataract; (**C**): unbalanced nostril structure, wherein the left nostril is higher than the right one; (**D**): finger injury; (**E**): punctiform scar in the upper lip.

**Figure 3 animals-13-01576-f003:**
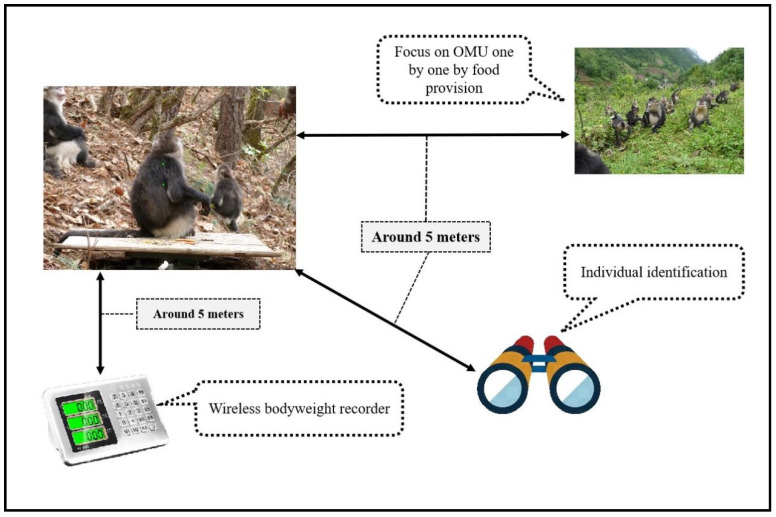
Measuring process for a given OMU or AMU. Before the monkeys traveled into the feeding site in the afternoon, a digital scale was placed on a relatively flat place and their favorite food was put on the scale. When a single individual walked on the scale, we recorded their body mass, once stable, together with identification information (age, gender, unit, etc.).

**Figure 4 animals-13-01576-f004:**
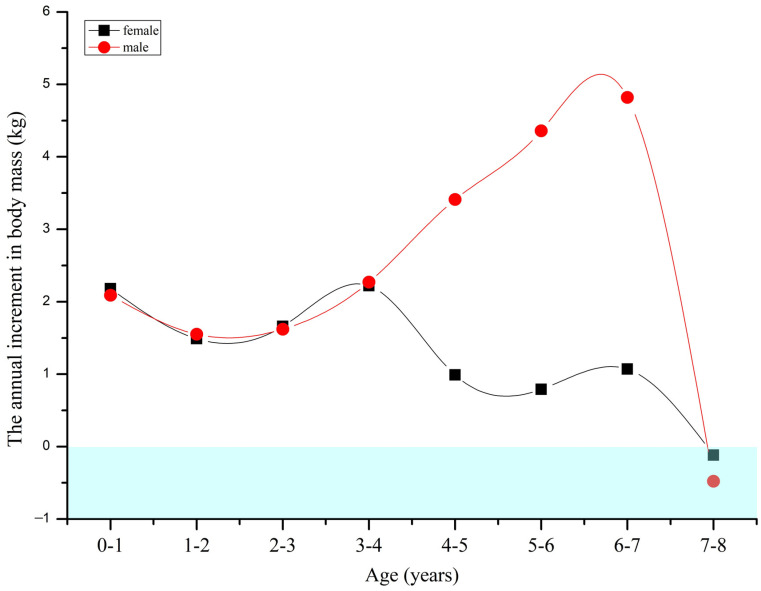
The body mass yearly increase of both sexes in *R. bieti.* The sample sizes were: birth to one-year group: F (female)/M (male) = 30/35; one to two-year group: F/M = 30/27; two to three-year group: F/M = 47/19; three to four-year group: F/M = 47/19; four to five-year group: F/M = 70/24; five to six-year group: F/M = 25/20; six to seven-year group: F/M = 25/20; seven to eight-year group: F/M = 112/35.

**Figure 5 animals-13-01576-f005:**
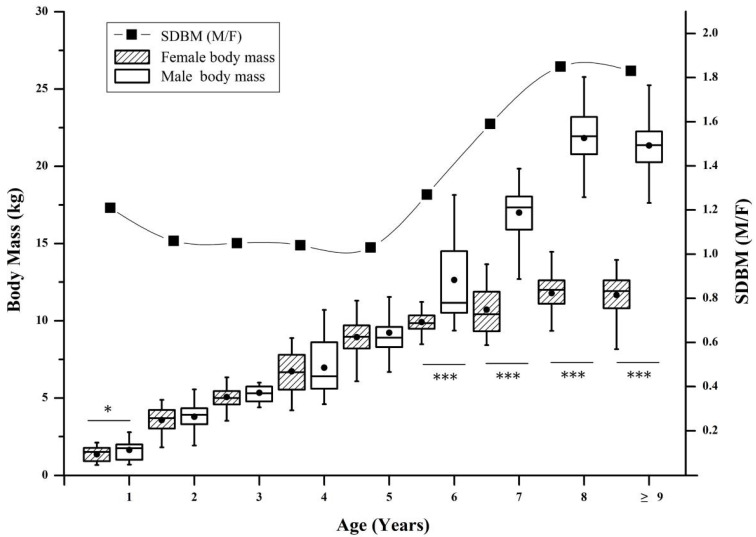
Ontogenetic development of sexual dimorphism in body mass of *R. bieti.* *: at *p* < 0.05; and ***: at *p* < 0.001. The sample sizes were: one-year group: F (female)/M (male) = 36/35; two-year group: F/M = 30/48; three-year group: F/M = 87/27; four-year group: F/M = 47/19; five-year group: F/M = 73/24; six-year group: F/M = 70/25; seven-year group: F/M = 25/20; eight-year group: F/M = 116/35; nine years or older group: F/M = 112/96.

**Figure 6 animals-13-01576-f006:**
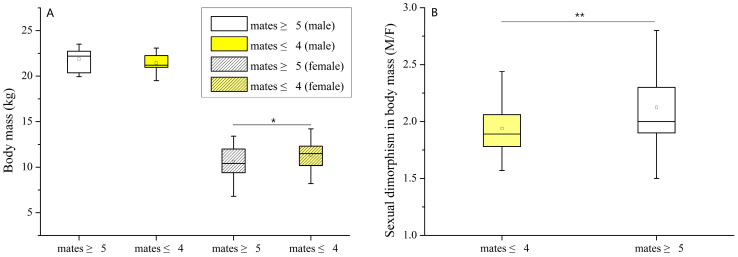
Body mass (**A**) and sexual dimorphism in body mass (**B**) comparison of the adults between the two OMU groups. *: at *p* < 0.05, **: at *p* < 0.01.

**Figure 7 animals-13-01576-f007:**
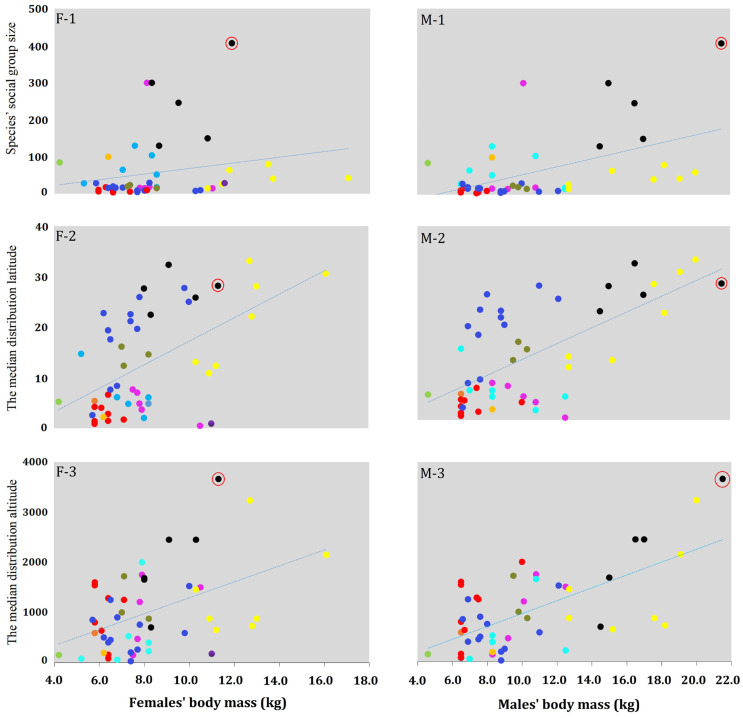
The relationship between adults’ body mass, species’ social group size, and distribution parameters among colobines. (**F-1**): between females’ body mass and species’ social group size; (**F-2**): between females’ body mass and median distribution latitude; (**F-3**): between females’ body mass and median distribution altitude; (**M-1**): between the males’ body mass and species’ social group size; (**M-2**): between males’ body mass and median distribution latitude; (**M-3**): between the males’ body mass and median distribution altitude. Each color repents a colobine genus: *Presbytis*: red, *Procolobus*: green, *Trachypithecus*: blue, *Piliocolobus*: cyan, *Colobus*: magenta, *Semnopithecus*: yellow, *Pygathrix*: dark yellow, *Rhinopithecus*: black, *Simias*: orange, *Nasalis*: purple. *R. bieti* was singled out by the red circle.

**Figure 8 animals-13-01576-f008:**
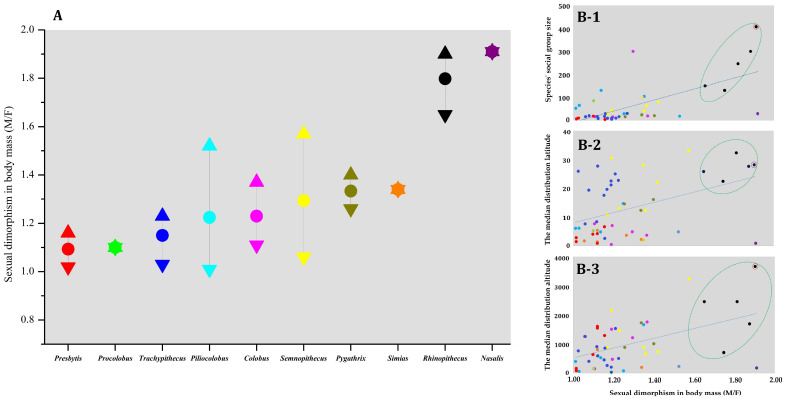
Sexual dimorphism in body mass (male/female, short in SDBM) and its relationship with species’ society group size and distribution parameters. (**A**): SDBM patterns of ten colobine genera, ▲: maximum value, ●: mean value, ▼: minimum value; (**B-1**): the relationship between SDBM and species’ society group size; (**B-2**): the relationship between SDBM and median distribution latitude; (**B-3**): the relationship between SDBM and median distribution altitude. Each color repents a genus (see Figure 7). The green circle represents the five species of the genus *Rhinopithecus*, and *R. bieti* is highlighted by the red circle.

**Figure 9 animals-13-01576-f009:**
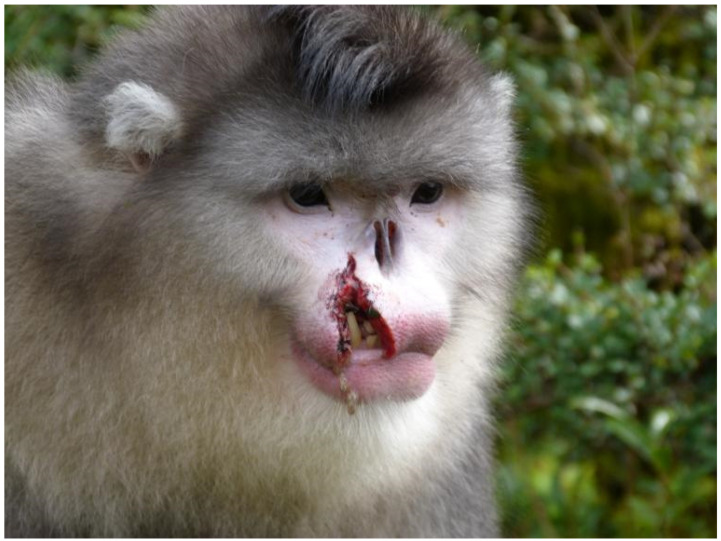
The image shows a severe injury of a male adult due to fierce fighting among males for reproductive opportunities.

**Table 1 animals-13-01576-t001:** Population information studied in Shangri-La Yunnan Golden Monkey National Park in 2010–2013.

Date	Unit	Number				
		Infant	Juvenile	Adult Female	Adult Male	Total
15 March 2013	DGZ OMU	2	3	4	1	10
	YDH OMU	1	4	4	1	10
	HC OMU	1	4	2	1	8
	PG OMU	1	1	1	1	4
	DB OMU	1	3	2	1	7
	BL OMU	0	2	2	1	5
	LHG OMU	0	4	2	1	7
	AMU	-	7	-	5	12
	Total	6	28	17	12	63
15 April 2012	DGZ OMU	3	3	6	1	13
	YDH OMU	1	4	4	1	10
	HC OMU	0	5	6	1	12
	BL OMU	1	4	3	1	9
	XW OMU	1	4	2	1	8
	DHZ OMU	2	2	7	1	12
	SHB OMU	1	0	1	1	3
	LHG OMU	1	4	5	1	11
	AMU	-	6	-	3	9
	Total	10	32	34	11	87
10 May 2011	DGZ OMU	1	4	4	1	10
	YDH OMU	2	4	4	1	11
	HC OMU	2	6	3	1	12
	XW OMU	0	6	5	1	12
	BZH OMU	0	3	3	1	7
	SHB OMU	0	3	1	1	5
	LHG OMU	2	5	3	1	11
	AMU	-	19	-	6	25
	Total	7	50	23	13	93
4 May 2010	DGZ OMU	1	4	5	1	11
	YDH OMU	2	4	4	1	11
	HC OMU	2	6	4	1	13
	XW OMU	4	6	5	1	16
	SM	3	3	3	1	10
	SHB OMU	0	3	2	1	6
	LHG OMU	1	5	3	1	10
	AMU	-	19	-	10	29
	Total	13	50	26	17	106

Note: Infants are less than one year old; juveniles are individuals older than one year old, but before sexual maturity; adult females are sexual matured individuals ≥ five years old; and adult males are sexual matured individuals ≥ eight years.

**Table 2 animals-13-01576-t002:** Adult male and female mean body mass, sexual dimorphism in body mass, social group size, and distribution parameters (median latitude and altitude) of colobines.

Species Name		Body Mass (kg)		SDBM	Society Group Size	Middle Altitude	Middle Latitude	Data Source
		Male	Female					
*Colobus*	*angolensis*	10.1	7.8	1.29	300	1208	4.8588	[38,54]
	*guereza*	10.8	7.9	1.37	19	1750	3.6718	[55,56]
	*polykomos*	8.3	7.5	1.11	16	145	7.6471	[55,54]
	*satanas*	12.5	10.5	1.19	15	1500	0.4689	[54,57]
	*vellerosus*	9.2	7.7	1.20	15	471	7.0174	[58]
	Average value	10.2	8.3	1.23	73	1015	4.7328	
*Piliocolobus*	*badius*	8.3	8.2	1.01	52	395	6.0829	[59]
	*kirkii*	7.0	6.8	1.03	65	55	6.1162	[60]
	*oustaleti*	12.5	8.2	1.52	18	225	4.8372	[54]
	*pennantii*	10.0	7.9	1.27	30	2000	3.6497	[54,61]
	*preussi*	8.3	7.3	1.14	130	525	4.8098	[62]
	*temminckii*	6.5	5.2	1.25	29	75	14.6684	[63,64]
	*tephrosceles*	10.8	8.0	1.34	104	1655	2.0079	[54]
	Average value	9.1	7.3	1.25	61	704	6.0246	
*Procolobus*	*verus*	4.6	4.2	1.10	85	150	5.2309	[55]
*Presbytis*	*canicrus*	6.5	5.8	1.13	12	800	1.3628	[65]
	*comata*	7.4	6.4	1.15	4	1283	6.6146	[65]
	*hosei*	6.5	5.8	1.13	6	1600	4.2013	[65]
	*melalophos*	7.5	7.1	1.06	6	1250	1.6919	[65]
	*potenziani*	6.5	6.4	1.02	6	157	2.8091	[65]
	*rubicunda*	6.5	5.8	1.13	11	1541	0.8511	[65]
	*sabana*	6.5	5.8	1.13	7	587	5.3748	[65,66]
	*siamensis*	6.3	6.5	0.96	18	635	4.0279	[67]
	*siberu*	6.5	6.4	1.02	6	80	1.4403	[54]
	Average value	6.5	6.3	1.03	8	881	3.1526	
*Semnopithecus*	*ajax*	20.0	12.7	1.57	60	3235	33.0853	[68]
	*entellus*	18.2	12.8	1.42	80	728	22.0778	[65]
	*hector*	17.6	13.0	1.35	41	875	28.0362	[69]
	*hypoleucos*	12.7	10.3	1.23	15	1455	13.0609	[65]
	*johnii*	12.7	10.9	1.17	27	875	10.8681	[65]
	*priam*	15.2	11.2	1.36	64	650	12.3023	[70]
	*schistaceus*	19.1	16.1	1.19	43	2150	30.5505	[71]
	*vetulus*	6.9	6.5	1.06	16	1250	7.6041	[65]
	Average value	15.3	11.7	1.29	43	1402	19.6982	
*Trachypithecus*	*crepusculus*	6.9	6.4	1.08	20	400	19.3330	[72]
	*cristatus*	6.6	5.7	1.16	29	850	2.5591	[54]
	*delacouri*	9.0	7.7	1.17	9	258	19.6117	[72]
	*francoisi*	8.0	7.8	1.03	10	755	25.9110	[72]
	*geei*	11.0	9.8	1.12	8	588	27.7097	[54]
	*hatinhensis*	7.5	6.5	1.15	17	450	17.5400	[72]
	*leucocephalus*	8.8	7.4	1.19	9	200	22.5104	[72]
	*obscurus*	7.6	6.8	1.12	17	900	8.3603	[54]
	*phayrei*	7.6	6.2	1.23	16	500	22.7374	[54]
	*pileatus*	12.1	10.0	1.21	10	1525	24.9788	[65,73]
	*poliocephalus*	8.8	7.4	1.19	5	25	21.1436	[54]
	Average value	8.5	7.4	1.15	14	586	19.3086	
*Pygathrix*	*cinerea*	10.3	8.2	1.25	16	874	14.5574	[54]
	*nemaeus*	9.8	7.0	1.40	20	1000	16.0972	[54]
	*nigripes*	9.5	7.1	1.34	24	1721	12.3170	[54]
	Average value	9.9	7.4	1.33	20	1198	14.3239	
*Rhinopithecus*	*bieti*	21.5	11.3	1.90	407	3663	28.1467	This study
	*brelichi*	15.0	8.0	1.88	300	1685	27.6115	[50]
	*roxellana*	16.5	9.1	1.81	246	2450	32.3089	[74]
	*avunculus*	14.5	8.3	1.75	130	700	22.4123	[50]
	*strykeri*	17.0	10.3	1.65	150	2450	25.8137	[75]
	Average value	16.9	9.4	1.80	247	2190	27.2586	
*Simias*	*concolor*	8.3	6.2	1.34	100	192	2.1931	[76,77]
*Nasalis*	*larvatus*	21.0	11.0	1.91	150	175	0.9012	[78]

Note: The geographic range of each species in colobines downloaded from IUCN red list.

## Data Availability

All data generated or analyzed during this study are included in this published article, and publicly available repositories.

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
