# Peer review of "Ontogenetic Development of Sexual Dimorphism in Body Mass of Wild Black-and-White Snub-Nosed Monkey (Rhinopithecus bieti)"

_animals, 2023, doi:10.3390/ani13091576_

Round 1
Reviewer 1 Report
Please get an accomplished English speaker to edit the text. It is very difficult to understand at the moment. Thank you.
Author Response
Dear reviewer,
Thank you for your suggestion, and so sorry for the manuscript was difficult to read due to our poor English, so that we invited Susan Olivier from UK have read and corrected English. The language of the manuscript has greatly improved.
Reviewer 2 Report
Thank you for the opportunity to review this article. I appreciate the many challenges involved in collecting the data, and I am excited by the complex evolutionary questions that can be answered by the longitudinal dataset available from this site. I enjoyed reading this manuscript.
The authors set out to describe and explain sexual dimorphism in body mass that occurs in Rhinopithecus bieti, a colobine monkey species with an extreme high-altitude distribution. They draw on an impressive and detailed data set spanning years of monitoring multiple groups of this species, and they compare their results to those reported for other colobines. A strength of the manuscript is the authors’ unique ability to analyze their data ontogenetically, so that the reader can clearly see the point in life history at which male and female body mass diverges—this provides rare clues to the evolutionary drivers of dimorphism in this species. Evolutionary pressures considered in the article include diet—the folivorous adaptations of this species are key to its exploitation of a resource-poor environment--and social organization (one male units with high numbers of adult females to adult males).
The figures are helpful and clear, but I note here that Figure 1, which I just loved, is a bit fuzzy at least in the view I had. I found the figure comparing various colobine taxa especially insightful and helpful in considering the evolutionary pressures that each colobine faced as they adapted to diverse habitats in Asia. Figure 2 is remarkable in the detail it provides for this little-known taxon that is very challenging to study in nature.
I recommend accepting the manuscript for publication following revision. I have three major suggestions and two minor ones for the authors to consider in the revision.
Major:
1. The monkeys at this site have been provisioned for many years. How might that fact impact on body mass? (Referred to on line 178). For other species, provisioning results in younger ages at maturation for females. Your data are interesting in that R. bieti females mature late—5 years—compared to taxa such as Trachypithecus obscurus and T. francoisi, where I am seeing online reports of 3-4 years and 4-5 years, respectively. I am thinking that unprovisioned R. bieti females would mature even later than what you report, which is interesting because it suggests to me a selective pressure exerted on both sexes to grow to a fairly large mass before taking distinct male/female paths. Exploring this idea a bit more would lend strength to your argument, I think, and I think would be linked to the species’ folivorous and seasonal diet.
2. Figure 5 is so interesting and clearly shows that divergence between males and females at year 5. I note, though, that the standard deviations are quite large and wonder about the appropriateness of using the average? I was quite intrigued by your reports of marked variation in body mass on an annual basis. I think this question/interest relates to my next point:
3. Seasonal fluctuation in body mass fascinates me and led me to wonder whether you would see less body mass dimorphism if data were used from particular seasons. My guess is that males’ body mass varies more seasonally than females’ does (although figure 5 does not seem to show that). This also led me to wonder whether the selective pressure for R. bieti is on females to stay relatively smaller, rather than for males to be bigger. This selective pressure might be particularly salient in nutrient-poor habitats such as this one, where overall large size is needed to access nutrients provided in a folivorous and seasonal diet, but where selection would favor females to become just that large and no larger, due to the increased energetic demands of lactation, gestation, and childcare (which I believe is distributed in this species) placed on them. So the question is not to ask why males are so large, but rather why females are not larger.
Minor:
1. Line 68 should be changed to read “some species of gibbon…” rather than most species.
2. R. bieti does appear to be the colobine species with this highest altitudinal distribution (for example, lines 92-93), but there are reports of cercopithecine species (Macaca munzala [2000-3000 m asl]and M. selai [>4000 m asl]) that are higher—please see citations below as examples. Your point that this is the highest colobine is important and the cross-colobine comparisons demonstrate this point very well!
Ghosh A, Thakur M, Singh SK, Dutta R, Sharma LK, Chandra K, Banerjee D. The Sela macaque (Macaca selai) is a distinct phylogenetic species that evolved from the Arunachal macaque following allopatric speciation. Mol Phylogenet Evol. 2022 Sep;174:107513. doi: 10.1016/j.ympev.2022.107513. Epub 2022 May 20. PMID: 35605928.
Sinha, A., Datta, A., Madhusudan, M.D. et al. Macaca munzala: A New Species from Western Arunachal Pradesh, Northeastern India*. Int J Primatol 26, 977–989 (2005). https://doi.org/10.1007/s10764-005-5333-3
Author Response
To reviewer 2,
Dear reviewer,
Thank you for your recognition of our research and suggestions for improving.
The authors set out to describe and explain sexual dimorphism in body mass that occurs in Rhinopithecus bieti, a colobine monkey species with an extreme high-altitude distribution. They draw on an impressive and detailed data set spanning years of monitoring multiple groups of this species, and they compare their results to those reported for other colobines. A strength of the manuscript is the authors’ unique ability to analyze their data ontogenetically, so that the reader can clearly see the point in life history at which male and female body mass diverges—this provides rare clues to the evolutionary drivers of dimorphism in this species. Evolutionary pressures considered in the article include diet—the frivolous adaptations of this species are key to its exploitation of a resource-poor environment--and social organization (one male units with high numbers of adult females to adult males).
Respond: Thank you for approving our job.
The figures are helpful and clear, but I note here that Figure 1, which I just loved, is a bit fuzzy at least in the view I had. I found the figure comparing various colobine taxa especially insightful and helpful in considering the evolutionary pressures that each colobine faced as they adapted to diverse habitats in Asia. Figure 2 is remarkable in the detail it provides for this little-known taxon that is very challenging to study in nature.
Respond: Thank you for approving our job, and we have redrawn the Figure 1 and improved its resolution.
I recommend accepting the manuscript for publication following revision. I have three major suggestions and two minor ones for the authors to consider in the revision.
Major:
- The monkeys at this site have been provisioned for many years. How might that fact impact on body mass? (Referred to on line 178). For other species, provisioning results in younger ages at maturation for females. Your data are interesting in that bietifemales mature late—5 years—compared to taxa such as Trachypithecus obscurus and T. francoisi, where I am seeing online reports of 3-4 years and 4-5 years, respectively. I am thinking that unprovisioned R. bieti females would mature even later than what you report, which is interesting because it suggests to me a selective pressure exerted on both sexes to grow to a fairly large mass before taking distinct male/female paths. Exploring this idea a bit more would lend strength to your argument, I think, and I think would be linked to the species’ folivorous and seasonal diet.
Respond: This work began in the first three years under feeding, we think the sex dimorphism in body mass do not undergo changes in such short time. We think the species’ folivorous and seasonal diet maybe are the key reasons for the extremely significant sexual dimorphism in body mass in R. bieti, but we do not have enough data and previous researches to support it, folivorous and seasonal diet due to the acquisition of energy is very difficult, the pressure of energy acquisition is also a major driving factor for sexual dimorphism, but we did not some work on these, this a good idea, we will do some work about it in the future.
I will also do some work about your suggestion that provisioning results in younger ages at maturation for females, you know that our team do some work about R. bieti in Lasha mountain where there are one or more natural population, and we have habituated one of populations without provisioning under long time (more than 10 years) high intensity tracking.
- Figure 5 is so interesting and clearly shows that divergence between males and females at year 5. I note, though, that the standard deviations are quite large and wonder about the appropriateness of using the average? I was quite intrigued by your reports of marked variation in body mass on an annual basis. I think this question/interest relates to my next point:
Respond: Thank you for your suggestion, we cannot keep tracking every monkey all the time, it is difficult to ensure weighed the same individual in every time. Although it is provisioning by human, but it is completely freedom and can leave population at any time. During our research period, many monkeys have left or joined our study population, so currently we can only report such studies based on averages.
Of course, we have tracked a few individuals all the time, they body mass development trend is consistent with the result in figure 5,and we reported is consistent with the situation we actually observed. I've been there for 4 years, over 300 days per year, unfortunately, we just got these data.
- Seasonal fluctuation in body mass fascinates me and led me to wonder whether you would see less body mass dimorphism if data were used from particular seasons. My guess is that males’ body mass varies more seasonally than females’ does (although figure 5 does not seem to show that). This also led me to wonder whether the selective pressure for R. bieti is on females to stay relatively smaller, rather than for males to be bigger. This selective pressure might be particularly salient in nutrient-poor habitats such as this one, where overall large size is needed to access nutrients provided in a folivorous and seasonal diet, but where selection would favor females to become just that large and no larger, due to the increased energetic demands of lactation, gestation, and childcare (which I believe is distributed in this species) placed on them. So the question is not to ask why males are so large, but rather why females are not larger.
Respond: Thank you, this is a good suggestion, we try to do same work use the same seasons data and the result was consistent with this work, we will try to do some work to answer that why females are not larger in R. bieti in the further, i can not answer it now.
Minor:
- Line 68 should be changed to read “some species of gibbon…” rather than most species.
Respond: Thank you, we have modified it.
- bietidoes appear to be the colobine species with this highest altitudinal distribution (for example, lines 92-93), but there are reports of cercopithecine species (Macaca munzala [2000-3000 m asl]and M. selai [>4000 m asl]) that are higher—please see citations below as examples. Your point that this is the highest colobine is important and the cross-colobine comparisons demonstrate this point very well!
Ghosh A, Thakur M, Singh SK, Dutta R, Sharma LK, Chandra K, Banerjee D. The Sela macaque (Macaca selai) is a distinct phylogenetic species that evolved from the Arunachal macaque following allopatric speciation. Mol Phylogenet Evol. 2022 Sep;174:107513. doi: 10.1016/j.ympev.2022.107513. Epub 2022 May 20. PMID: 35605928.
Sinha, A., Datta, A., Madhusudan, M.D. et al. Macaca munzala: A New Species from Western Arunachal Pradesh, Northeastern India*. Int J Primatol 26, 977–989 (2005). https://doi.org/10.1007/s10764-005-5333-3
Respond: the highest distribution altitude of R. bieti is more 4000 m, the highest altitude is near 5000 m of the north populations in the north part of Yunnan and Tibet, and almost of the natural populations are distribute altitude should be higher than 3000 m, you know the dark coniferous forest is their main natural habitat.

Reviewer 3 Report
The aim of the manuscript is to quantify the level of sexual dimorphism in body mass in a species of sub-nosed monkey (Rhinopithecus bieti), test the influence of group size and compare it in other colobine species. This work is conducted on observations acquired in the field, on wild populations.
The paper is interesting and really into the aims and scopes of the journal. Many authors have published on sexual dimorphism in body mass in primates, but this manuscript is original in the sense that it focuses on a species of Chinese primates for which the knowledge is score and that doesn’t seem to follow the tendency drown by the other species of the group, probably because of its unusual habitat and ecology.
The manuscript suffers of many typos, forms errors and unclear sentences and really needs proofreading and corrections before acceptance. Some figures need to be reworked… I have a lot of comments on the pdf file that could help the authors in that way.
On the substance of the paper, the hypotheses are well expressed, the analyses are well performed, but the paper need some details. For instance, the authors announce that they’ll analyse the result to identify two macroevolutionary rules such as Bergmann and Rensch’s rules but actually they just mentioned these rules (never explain them). and didn’t really test such effect. The analysis need to be details in term of statistics, better explain and they authors need to remove all the confusions that could occur by reding the manuscript.
The discussion is week and lack many references. Several points are discussed, but really need to be deeply investigated.
For those reasons, I think that the manuscript deserves to be reworked but will, I am really sure, constitute an accurate and rigorous article publishable in the journal Animals.

Author Response
To reviewer 3,
Dear reviewer,
Thank you for your recognition of our research and suggestions for improving. Your extensive work and suggestions have greatly improved our manuscript, we invited Susan Olivier from UK have read and corrected English, the language of the manuscript has greatly improved.
We have modified the article one by one under your comments on the .pdf file.
Kind regards,
